# Using Annealing Treatment on Fabrication Ionic Liquid-Based PVDF Films

**Yung Ting \***, **Suprapto**, **Naveen Bunekar, Kulandaivel Sivasankar** and **Yopan Rahmad Aldori**

Mechanical Engineering Department, Chung Yuan Christian University, Chung li, Taoyuan 320,
Taiwan; praptomeunimed@gmail.com (S.); naveenbunekar@gmail.com (N.B.);
sivasankarmpm@gmail.com (K.S.); yopanaldori11@gmail.com (Y.R.A.)

**\*** Correspondence: yung@cycu.edu.tw; Tel.: +886-936-345-149

**Abstract:** In this study, a simple method to obtain pure β-phase directly from the melt process is proposed. A series of PVDF and ionic liquid (IL) was prepared by a solvent casting method with appropriate associated with the subsequent annealing treatment. IL plays a role of filler, which can create strong electrostatic interaction with PVDF matrix and directly induce β-phase crystallization on the PVDF during the melt. PVDF film sample is immersed in hot water for annealing treatment at different temperatures (25 °C to 70 °C). We found that annealing in high temperatures especially can not only increase more IL inserted into the amorphous region of polymer matrix to make more phase transformation, but also accelerate IL removal. Characteristics and performance of the PVDF films were investigated by use of FTIR, XRD, SEM, and AFM. Piezoelectric coefficient $d_{33}$ as well as $d_{31}$, degree of crystallinity, and sensitivity are measured in experiment to verify the performance of PVDF film.

**Keywords:** PVDF thin film; DMF; ionic liquid; solution casting; piezoelectric properties

---

## 1. Introduction

PVDF as a smart material has been widely used for miscellaneous application fields, such as sensor and actuator, energy harvesting, and medical devices due to the merits of flexibility, light weight, high sensitivity, excellent response over a wide frequency, and so on [1–8]. Compared to PZT ceramics, PVDF does not have the negative defects of brittleness, large stiffness, weight, and thermal conductivity, etc. [9]. The α-phase is directly obtained from the melt in the form of a compact film, which gives good mechanical properties and thermal stability but poor piezoelectricity [10,11]. A large fraction of β-phase implies more successful transformation from α to β phase, which is the goal, and significant performance index of gaining better piezoelectricity between the two piezoelectric crystalline (β and γ) phases of PVDF [12,13]. To produce large γ-phase, content has to use thermal treatment at extremely high temperature around 225 °C and 300 MPa, which is higher than the operating in this article [11,14]. Hence, the process of producing a large amount of β-phase introduced in this study is unlikely to have a significant fraction of γ-phase. How to obtain high percentage of β -phase and electromechanical properties is always of great concern for most of the sensor application. A few previous studies used a sheet of commercial raw PVDF film to carry out the later stretching and poling processes. In search of better stretching ratio, temperature, speed and direction in the process of stretching, and employment of higher voltage in the process of poling are probably the common ways to increase the content of β-phase and improve the quality [15–18].

Recently, many researchers have developed inorganic copolymer piezoelectric materials such as poly(vinylidene fluoride-trifluoroethylene) [P(VDF-TrFE)], chlorotrifluoroethylene [P(VDF-CTFE)], and hexafluoropropylene [P(VDF-CTFE)], which are excellent candidates of piezoelectric and dielectric

materials for a micro-electro mechanical system [MEMS] [19], microsensor [20], nanogenerator [21], and energy harvesting [22]. But the copolymer is quite expensive [23]. Instead, an aim in this study is to fabricate the PVDF film by means of solution casting method. Although the PVDF film is prepared by solvent casting and by evaporation at room temperature, it is unlikely to directly obtain enough content of β-phase from the melt. Also, a porous and fragile phenomenon occurs that would degrade the electromechanical properties and make the subsequent poling process infeasible [24]. Therefore, searching for appropriate solvent and further treatment in order to preserve good piezoelectricity is demanding. As known, α-phase is the only result after casting the PVDF films in solution has *trans-gauch* conformation (TGTG). If the solution is prepared at 70–90 °C by using DMF as solvent, the viscosity of the solution is decreased, and the thermal energy is strong enough to rotate the $CF_2$ groups through large-scale conformational change of the trans planar zigzag conformation (TTT). As a result, β-phase is formed [25].

Ionic liquid (IL) is a salt-like material with good properties of ion conductivity that appears to be liquid at room temperature [26]. Because of its nano-structure with high conductivity, low volatility, and good thermal, electrochemical, and chemical stability, IL has the advantage over traditional solvents and has become widely used in various applications in industry [27]. The IL-based polymers matrix has high conductivity ($10^{-4}$ to $10^{-2}$ S cm$^{-1}$) and high electrochemical stability (4–5.7 V) and thermostability (up to 300 °C) can be used for lithium-ion batteries and other solid electrolyte, dye-sensitized fuel cells and supercapacitors [28–30]. IL/PVDF composites have been developed for electrolyte in electromechanical actuator based in carbon nanotubes [30,31]. IL-polymer have been successfully developed in biosensors, optical sensors, and conducting polymer sensors [31,32]. On the other hand, IL based on quartz crystal polymer and piezoelectric substrate have been investigated for surface acoustic wave (SAW) with a frequency range of 10–30 MHz [32,33].

Using IL as a filler is an alternative method can produce pure β-PVDF directly from the melt due to its strong electrostatic interaction upon melt crystallization. Thus, IL easily interacts with the PVDF chains and effectively promoting the β-phase content [34]. Hence, an attempt to increase β-phase content is proposed by the use of DMF as a solvent to form the PVDF matrix (PVDF/DMF), and IL as a filler to support strong interaction with the dipolar moment of PVDF. Moreover, the PVDF film is immersed in hot water for annealing treatment after the processes of casting, evaporation, and air dry at room temperature. As we found, the mobility of ion increased, which facilitates the ion-dipole to easily interact between the $CF_2$ of PVDF and the imidazolium cation of IL [35,36].

A few techniques have investigated the ability of controlling and improving the content of β and γ phases of PVDF [11,14,37]. As known, IL is easy to melt below temperatures of 100 °C, accompanied with large quantities of cations and anions. While immersing the PVDF film into hot water, ion mobility is increased that would facilitate the interaction with the molecules and hydrogen bonding on the contact surface of PVDF film. Therefore, after solution casting, annealing treatment is proposed not only to wash out IL, but also provide an environment of increasing the interaction between the PVDF chains and ion liquid. As a result, part of IL is inserted into the amorphous region of polymer matrix and nonpolar crystal in PVDF, which can induce the PVDF chain to fold or transfer into polar phase.

In this article, a composite film based on PVDF materials with ionic liquids (IL) as filler to directly obtain pure β-phase from the melt is developed with good results. The solution casting method associated with appropriate ratio of DMF and IL as well as annealing treatment is proposed as a fast and simple way of producing high-quality PVDF thin film with a large fraction of β phase content, satisfactory for sensor and actuator applications for instance. Temperature control of the hot water in which PVDF is immersed for annealing treatment is also investigated and addressed with more detailed observations.

## 2. Materials and Methods

### 2.1. Materials

The commercial PVDF pellets (PVDF, Kynar, 740, Arkema in Taiwan) have a molecular weight ($M_w$: 180.000, Tg: −40 °C and Tm: 168 °C), specific gravity: 1.78 g/cc, melt flow: 1.1 g/10 min, and an average pellet particle size of 5 mm. The N, N-dimethylformamide (DMF) contains 99.5% of $C_3H_7NO$ and the ion liquid (IL) is made of $C_6H_{11}BF_4N_2$ with stated purity of 98%.

### 2.2. Fabrication of the PVDF Films Composite

The fabrication process of PVDF film by use of solvent casting method is illustrated in Figure 1. Firstly, the PVDF pellets was dissolved in the solvent DMF (20 wt% concentration). Then, the IL (10 wt%) was added and dissolved in PVDF/DMF solution under magnetic stirring at 80 °C overnight in order to make homogeneous mixture. Afterwards, the mixture (PVDF/DMF+IL), named with the abbreviation PDI, was casted on a glass support using a casting knife of 300 μm working thickness. The casted membrane film was placed at 90 °C in a vacuum oven for solvent evaporation overnight to remove the solvent. Then, the film was cooled down to 35 °C for 30 min, and then left at room temperature. Secondly, the film was immersed in a water bath to remove excessive solvent IL at various temperatures (25, 40, 50, 60, and 70 °C) for 2 h, and then removed moisture by air dry. Without operating the above second step, i.e., PDI composite film without immersion in hot water for annealing treatment (AT), named with the abbreviation PDI no_AT, was prepared for the purpose of comparison. A few pieces of the fabricated PVDF films with thickness of ±25 μm and area of $10 \times 15$ mm$^2$ were selected to carry out the following characterization tests.

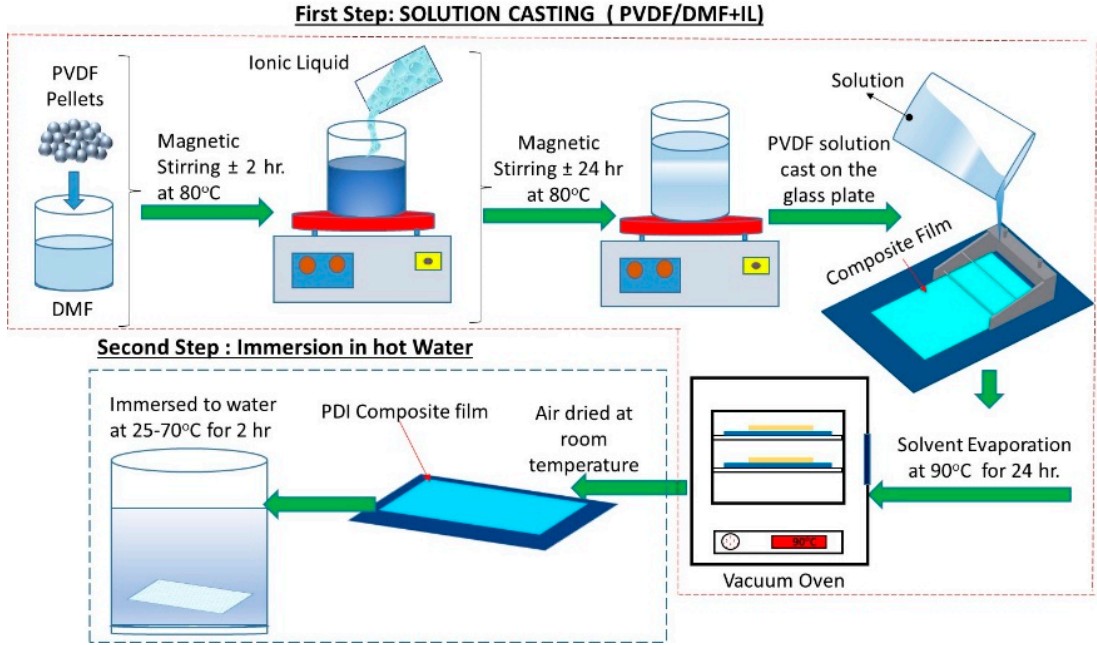

**Figure 1.** Schematic diagram of PVDF fabrication process.

### 2.3. Measurement and Characterization

A series of examinations of the PVDF film made by PVDF/DMF without/with IL filler were carried out for comparison. The thickness of the PDI composite film were measured by digitial thickness gauges (Elcometer 456 model EF1, London, Britain). Thickness of the made PVDF films were around 23.9–25.9 μm, suitable for the sensor application purpose for example [38]. Fourier transform infrared spectroscopy (FTIR) measurements were performed at room temperature

using a FTIR (Jasco FT/IR-4200, Tokyo, Japan) spectrometer apparatus with resolution of 4 cm$^{-1}$ was used to observe the crystallinity changes of the PVDF samples and estimate the fraction of β-phase in spectra bands in the range of 600–1600 cm$^{-1}$. X-ray diffraction (XRD: D8 Advance Eco, Brucker, Billerica, MA, USA) measurements were carried out by using Cu-Kα radiation for X-Ray with a wavelength of 1.54 Å used to characterize the crystal structure and estimate the degree of crystallinity. The surface morphology, roughness, and microstructure change of PVDF composites were examined by scanning electron microscopy (SEM, JEOL-JSM-7600F, Akishima, Japan) and atomic force microscopy (AFM-SPA-400/NanoNavi by Seiko Instruments, Saitama, Japan). The SEM measurements were carried out with a magnification range of 2000 × to 50,000 × at room temperature and observed on the condition of 10 kV acceleration voltage. The PVDF film samples were coated with a layer of Platinum (Pt) with thickness around 2–3 nm for 30 s at a pressure of ~20 mA in advance in order to enhance the electronic conductivity by sputter coater (JFC-1600 auto fine coater, JEOL, Tokyo, Japan). Surface image morphology and roughness were carried out by AFM with scanned area of 5 × 5 μm$^2$ and pixel resolution of 512 × 512 points was used to observe the surface topography and roughness of PDI composite films. The contact frequency of cantilever of 285 kHz with spring constant of 26 N/m was used during the measurements.

### 2.4. Measurement of the Piezolectricity

The output voltage response of $d_{31}$ and $d_{33}$ was measured by means of the extraction test method and impact force test method based upon the principle of energy conservation, respectively. Piezoelectric coefficient $d_{33}$ and $d_{31}$ of the PVDF composite films were measured by $d_{33}$ m and calculated by using extraction test, respectively. The output voltage response and sensitivity as well as piezoelectric coefficient are important indices, although without poling. The output voltage response of $d_{33}$ and $d_{31}$ were tested by means of an applied impact force as well as extraction force onto the PVDF film covered by silicon sheet on both sides to protect from damages. The measurement system is shown schematically in Figure 2a,b.

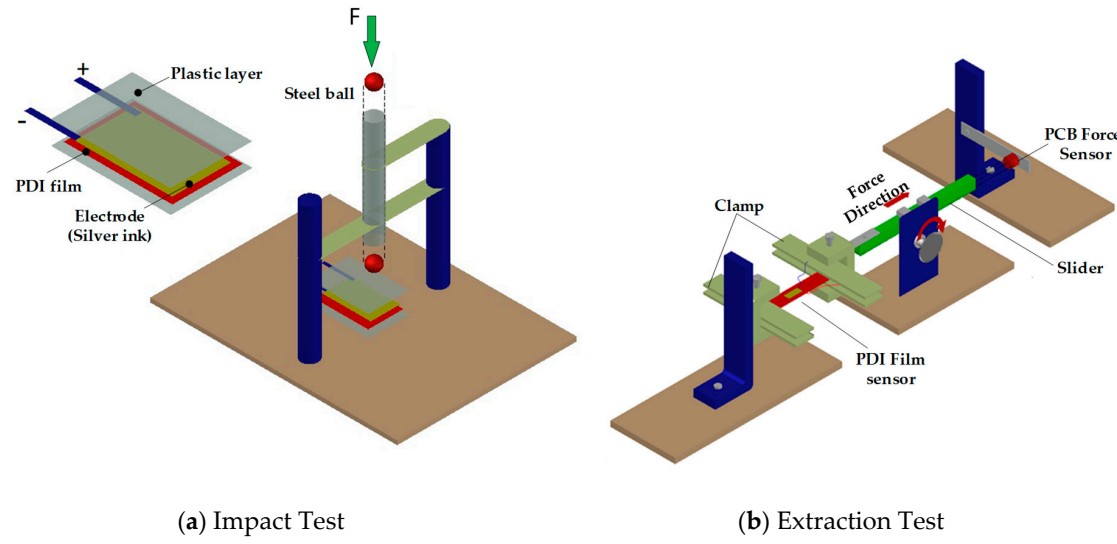

(**a**) Impact Test　　　　　　　　　　　　(**b**) Extraction Test

**Figure 2.** Test setup for measurement of piezoelectric coefficient (**a**) $d_{33}$, (**b**) ($d_{31}$) [39].

Impact force is generated by dropping a steel ball with four different mass (5, 8.3, 23.7, and 43.9 g with respect to diameters of 6.35, 12.7, 18, and 22 mm) at the height *h* of 150 mm. On the assumption of ideal transformation between potential energy (*mgh*) and kinetic energy, the steel ball, which makes contact with the PVDF film with kinetic energy, would generate an equivalent impact force *F* together with a bouncing back height *l*. Hence, the impact force is calculated to be 7.25, 12.20, 34.83, and 64.24 N,

respectively [39–41]. The extraction test was carried out by applying a stretch force on the PVDF film clamped at both ends [39].

## 3. Results and Discussion

### 3.1. FTIR Analysis to β-Phase Fraction

FTIR measurement was used to identify the characteristic bands and to confirm the crystalline phase of the PVDF sample. Figure 3 presents the FTIR spectra of PVDF films that were prepared after immersion in water at different temperature for 2 h. The corresponding FTIR spectra show all the absorption peaks of both the α and β phase in the range of 650–1600 cm$^{-1}$. In the FTIR spectra, the characteristic absorption bands of the crystalline α and β phase are labeled by an arrow for easy identification. According to the previous studies, the absorption peaks of α phase were located at 410, 489, 532, 614, 763, 795, 875, 1,166, 1,209, 1,401, and 1,423 cm$^{-1}$; the strong peaks of β-phase were located at 840 and 1,279 cm$^{-1}$; and the peaks of γ-phase were located at 811 and 1234 cm$^{-1}$ [14,42,43]. In Figure 3, the effect of immersion in water at different temperatures can be observed from the intensity of the peak of β phase. It is seen that a noticeable increase of intensity appears in the β-phase peaks of 838 cm$^{-1}$ and 1,072 cm$^{-1}$, respectively.

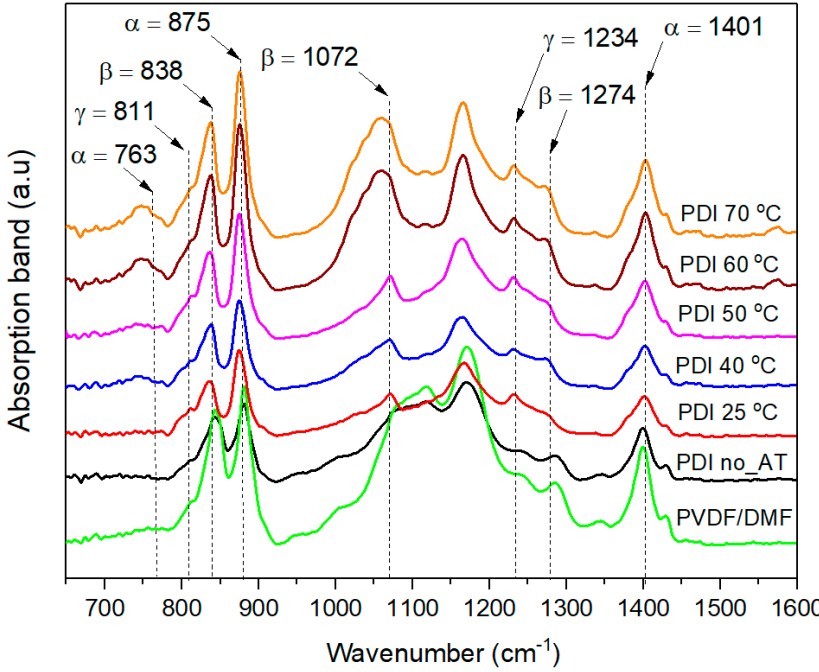

**Figure 3.** FTIR spectra without/with ionic liquid (IL) and annealing treatment at different temperatures.

It was observed that the characteristic of absorption bands at 1,072 cm$^{-1}$ was a little shifted by increasing the annealing temperature at 60 and 70 °C. Annealing temperature is one of the important factors that significantly influences the crystallize size and growth. During the temperature increase from room temperature to 70 °C, crystallinity in PVDF may change and cause the peak position of PVDF film to be shifted. In addition, the energy generated by the hot water is enough to rotate CF$_2$ dipoles and enhance the chain mobility, which produces more γ-phase and less fraction as well as instability of β-phase [42]. It implies that hot water for thermal annealing treatment plays a role of ignition source instrumental to stimulate the ionic conductivity and speed up the crystallization process as well as increase polar phase of crystallization for PVDF [36,44]. Moreover, it was found that more intensity happens in the case of water temperature at 70 °C. In addition, intensity of the γ-phase at 811 and 1234 cm$^{-1}$ is observed with a slight peak indicating transformation from part of the α-phase [34]. The various absorption band of PDI composites film are assigned and summarized in Table 1.

**Table 1.** FTIR analysis of PVDF/N, N-dimethylformamide (DMF)+IL (PDI) composite film.

| Wave Number (cm$^{-1}$) | Crystalline Phase | Group/Vibration |
|---|---|---|
| 763 | $\alpha$ | CF$_2$ rocking or In-plane bending [45] |
| 811 | $\gamma$ | HC out-of plane [46] |
| 838 | $\beta + \gamma$ | CH$_2$ rocking vibration [47] |
| 875 | $\alpha$ | CF$_2$ Asymmetric stretching [48] |
| 1072 | $\beta$ | CH$_3$ rocking [34,49] |
| 1234 | $\beta + \gamma$ | CF$_2$ Asymmetric stretching [14,34] |
| 1274 | $\beta$ | CF$_2$ Antisymmetric stretching [14] |
| 1401 | $\alpha$ | CH$_2$ wagging [34] |

The fraction of the β-phase crystallization on the samples is calculated by [50].

$$F(\beta) = \frac{A_\beta}{(k_\beta/k_\alpha)A_\alpha + A_\beta} = \frac{A_\beta}{(1.26)A_\alpha + A_\beta} \tag{1}$$

where the absorption coefficients are $k_\alpha = 6.1 \times 10^4$ and $k_\beta = 7.7 \times 10^4$ cm$^2$/mol; the value of $A_\alpha = 0.003537$ and $A_\beta = 0.057868$ are the absorbance bands corresponding to the two peaks at 763 and 838 cm$^{-1}$ from the Lambert–Beer law [50].

The calculated fraction of β-phase for all the cases are shown in Figure 4 with expression of error bars identified as the upper and lower 95% confidence intervals. As seen, hot water acts as a heat source that transmits energy to the PVDF film to enhance the formation of β-phase crystallization. The fraction of β-phase is increased as the temperature for annealing is increased, which implies temperature of hot water is a key factor influential to the content of β-phase [36]. The greatest β-phase fraction of 92.4% was achieved when the PVDF films were immersed in hot water at annealing temperature of 70 °C. It is noted that polar β-phase predominantly occurs when temperature falls below 70 °C. It is not suitable to allow temperature over 70 °C since a mixture of α and β phase exists between 70 °C and 110 °C and only α phase appears when temperature rises above 110 °C [50,51].

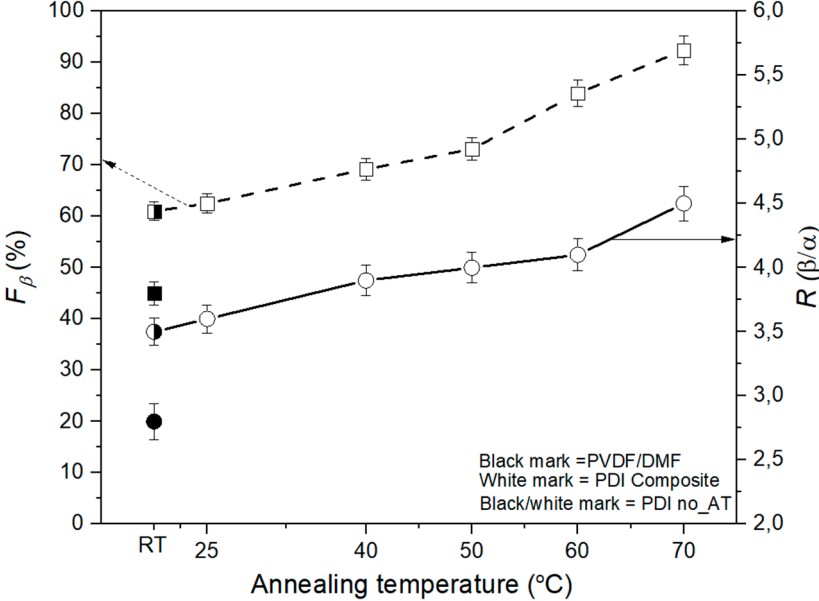

**Figure 4.** Fraction of β-phase $F^\circledR$ and ratio $R$ (β/α) vs. annealing temperature.

As mentioned above, the FTIR spectra of PVDF indicate distinct peaks for α and β phases at 763 cm$^{-1}$ and 838 cm$^{-1}$, respectively. The progress of phase transformation can be observed by the ratio of the absorbance band peaks of β to α phases, which is defined by [52]

$$Ratio\ R(\beta/\alpha) = \frac{Absorbance\ bands\ (A_\beta)}{Absorbance\ bands\ (A_\alpha)} \tag{2}$$

While the ratio is less than 1, the film is predominantly with α phase; on the contrary, the film is predominantly with β-phase for ratio is greater than 1. In Figure 4, all of the case studies imply a large amount of β-phase is gained for ration greater than 3.5. Annealing treatment in hot water at 70 °C would obtain the largest fraction of β-phase.

### 3.2. XRD Diffraction Pattern and Degree of Crystallinity

Figure 5 shows the X-ray diffraction measurement with the same PVDF samples of PVDF/DMF and PDI composite (PVDF/DMF+IL) immersion into hot water for annealing treatment. The case of PDI composite without annealing treatment (PDI no_AT) is also presented for comparison. The measurement results are listed in Table 2, which are close to those addressed in the previous studies [34,44]. For PVDF/DMF, the peaks appear at 2θ angle of 17.9°, 18.54°, 20.12°, and 26.6°. Three characteristic peaks at 2θ angles of 17.9°, 18.54°, and 26.6° corresponding to the crystal plane (100), (020), (021) respectively is attributed to the crystalline α-phase [14]. As observed in all of the PDI composite films, the α-phase peaks at 2θ angle appear at 17.6°, 26.29°, and 40.37° are attributed to the α-phase crystal plane (100), (021), and (002), respectively; the β-phase peak at 2θ angle is 20.26° attributed to both the β-phase crystal planes (110) and (200); and the γ-phase peak at 2θ angle is 18.5° attributed to the γ-phase crystal plane (020) [34].

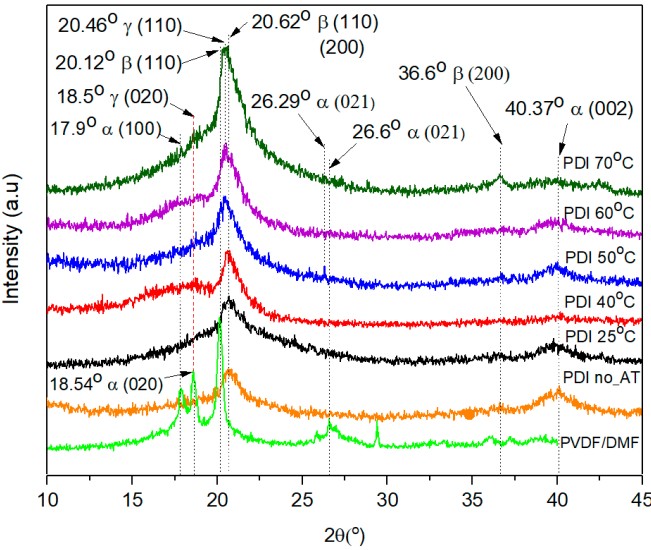

**Figure 5.** X-ray diffraction pattern of PVDF samples at different water temperatures.

The intensity of the β-phase and γ-phase peaks are significantly increased while increasing the hot water temperature for annealing treatment. On the other hand, the intensity of α-phase peak is diminished. In particular, a very strong diffraction peak was found at 2θ angle of 20.62° corresponding to the β-phase crystal planes (110) (200) [53–55]. As seen, a large fraction of β-phase is gained when increasing the immersion water temperature to 70 °C. In a previous study, the 2θ angle of β-phase on crystal planes (110) (200) and γ-phase on crystal plane (110) was 20.26° and 20.04°, respectively [14,44]. Since both locations are very close, it is difficult to distinguish between them. However, the FTIR spectroscopy shown in Figure 3 implies the content of γ-phase is not so dominant as the β-phase.

Table 2. Diffraction crystal plane of PVDF/DMF and PDI composite film.

| Sample | Present Study | | | | | | Previous Work Ref. [14,51,53,56–58] | | | | | |
| --- | --- | --- | --- | --- | --- | --- | --- | --- | --- | --- | --- | --- |
| | α | | β | | γ | | α | | β | | γ | |
| | 2θ (°) | Crystal Plane | 2θ(°) | Crystal Plane | 2θ(°) | Crystal Plane | 2θ(°) | Crystal Plane | 2θ(°) | Crystal Plane | 2θ(°) | Crystal Plane |
| PVDF/DMF | 17.9<br>18.54<br>26.6 | (100)<br>(020)<br>(021) | 20.12 | (110) | 18.5 | (020) | 17.66<br>18.3 | (100)<br>(020) | 20.6<br>20.26 | (110/200) | 18.30<br>19.20 | (020)<br>(002) |
| PDI no AT | 17.7<br>26.6<br>40.37 | (100)<br>(021)<br>(002) | 20.28 | (110)/(200) | 18.22 | (020) | 19.9<br>26.56<br>27.8 | (110)<br>(021)<br>(111) | 20.7<br>20.8<br>20.9 | (200)<br>(110) (200)<br>(110) (200) | 20.0<br>20.04<br>20.03 | (110)<br>(110)<br>(110) |
| PDI Composite 25 °C | 17.6<br>26.29<br>40.37 | (100)<br>(021)<br>(002) | 20.62 | (110)/200 | 18.17<br>20.3 | (020)<br>(110) | 35.7<br>39<br>41.1 | (200)<br>(002)<br>(111) | 36.6 | | 20.3<br>39.0 | (020)<br>(211) |
| 40 °C | 17.49<br>26.26<br>40.39 | (100)<br>(021)<br>(002) | 20.60 | (110)/(200) | 18.4<br>20.2 | (020)<br>(110) | | | | | | |
| 50 °C | 17.5<br>26.24<br>40.39 | (100)<br>(021)<br>(002) | 20.62 | (110)/(200) | 18.8<br>20.19 | (020)<br>(110) | | | | | | |
| 60 °C | 17.5<br>26.29<br>40.15 | (100)<br>(021)<br>(002) | 20.60 | (110)/(200) | 18.73<br>20.40 | (020)<br>(110) | | | | | | |
| 70 °C | 17.62<br>26.14<br>40.37 | (100)<br>(021)<br>(002) | 20.62<br>36.6 | (110)/(200)<br>(200) | 18.20<br>20.46 | (020)<br>(110) | | | | | | |

The degree of crystallinity (Xc) was calculated as a percentage of the integrated area of all crystalline peaks to the total integrated area under the XRD peaks. The degree of crystallinity is given by [59]

$$\text{Degree of crystallinity } (X_c) \ = \ \frac{S_c \ (crystalline \ phase)}{S_c \ (crystalline \ phase) + S_a \ (Amorphous \ phase)} \times 100\% \qquad (3)$$

where $S_c$ represents area of the peak corresponding to the crystalline α, β, and γ in crystalline phase; $S_a$ represents the area corresponding to the amorphous phase fraction. The results listed in Table 3 are smoothed by using a curve fitting method and expressed by a Gaussian function [39,60]. It was found that the maximum crystalline content in the PVDF film is about 46%–53.15% in the process of various thermal annealing conditions. As can be seen, the degree of crystallinity increased when the immersion temperature increased. In particular, the degree of crystallinity of PVDF film reaches the highest value for the case of immersion water temperature of 70 °C.

**Table 3.** Degree of crystallinity and ratio of β/α phase.

| Sample | Crystallinity Xc (%) | | Ratio (β/α) | |
|---|---|---|---|---|
| | **Present** | **Ref. [24,54,61]** | **Present** | **Ref. [54,55]** |
| PVDF/DMF | 40 | 33 | 3.3 | 2.12 |
| PDI no_AT | 42 | | | |
| PDI composite | | | | |
| −25 °C | 46.96 | | 3.6 | |
| −40 °C | 47.79 | | 3.9 | |
| −50 °C | 47.8 | 37, 41, 48, 54.5, 55.9, 56 | 4 | 2.7, 3.24, 3.7, 5.8 |
| −60 °C | 51.24 | | 4.1 | |
| −70 °C | 53.15 | | 4.7 | |

The results of FTIR and XRD measurement indicate that IL can induce the polar phase to obtain a great number of nuclei. Hence, it facilitates the transformation from α-phase to polar phase via the interaction between > CF2 groups and cations of IL [34,62]. Also, the annealing treatment would facilitate the PVDF chains in amorphous area to entirely flop and form more polar crystals [44,62]. To sum up, more β-phase transformation happens, attributed to activation of the ionic liquid and energy provision from the hot water to accelerate the reaction between solute and solvent. The immersion water temperature, which is an essential factor of annealing treatment, would affect the fraction of β-phase.

### 3.3. Output Voltage Response

Figure 6 shows the output voltage response is measured and presented with an average by a series of 10 times testing. Employed with the above four different impact force onto the PVDF film was carried out for searching $d_{33}$. The output voltage response for each fabricated PVDF film at different annealing water temperatures with expression of error bars is shown in Figure 6a. The average output voltage response versus impact force is shown in Figure 6b. Similarly, different annealing water temperature and stretch force applied to the film for searching $d_{31}$ are shown in Figure 7a,b, respectively.

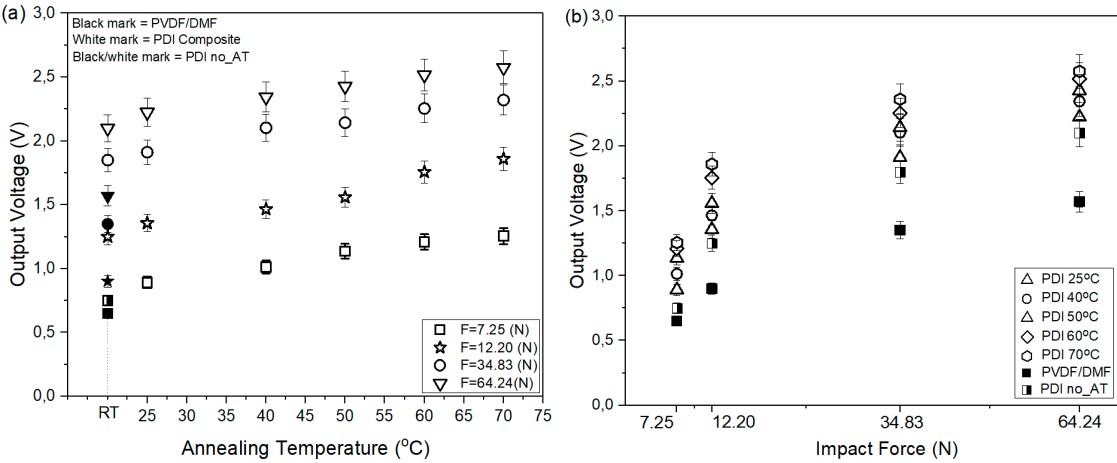

**Figure 6.** Output voltage response ($d_{33}$) vs. impact force and annealing temperature. (**a**) The output voltage response for each fabricated PVDF film at different annealing water temperatures with expression of error bars; (**b**) the average output voltage response versus impact force.

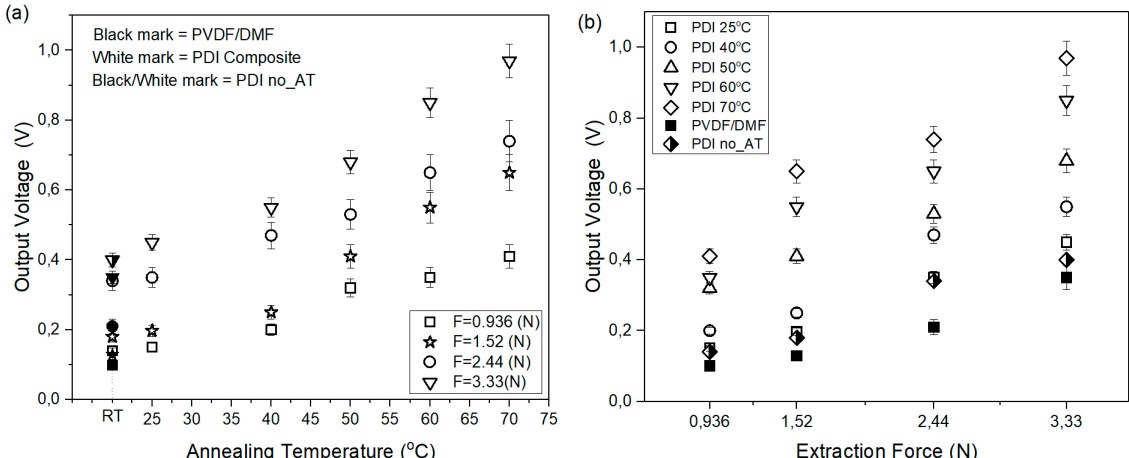

**Figure 7.** Output voltage response ($d_{31}$) vs. extraction force and annealing temperature. (**a**) Different annealing water temperature applied to the film for searching d31; (**b**) different stretch force applied to the film for searching d31.

### 3.4. Sensitivity of PVDF Sensor

Sensitivity and accuracy are the critical factors of a PVDF film sensor that significantly influence its performance [63,64]. The static sensitivity *S* is defined as the ratio of the output voltage (*V*) to the input impact force *F* acting on the PVDF film and calculated by [65].

$$S = \frac{Output\ Voltage}{Input\ Force} = \frac{V}{F}\left(\frac{Volt}{Newton}\right) \tag{4}$$

From the above measurement data in Figures 6 and 7, sensitivity of $d_{33}$ and $d_{31}$ of each type of PVDF film with expression of error bars is shown in Figures 8 and 9, respectively. As seen in Figure 8a, sensitivity is increased when the annealing water temperature is increased. The maximum sensitivity is about ± 0.20 V/N for the PDI composite film under annealing temperature of 70 °C. In Figure 8b, it is interesting to observe that different impact load on the PVDF films would affect the sensitivity. That is, low impact load applied on the PVDF would preserve good sensitivity but an impact load too large would reduce sensitivity significantly, which provides an index in application. Similarly, sensitivity of $d_{31}$ with respect to annealing temperature and extraction force are presented in

Figure 9a,b, respectively. As expected in general use of PVDF, sensitivity $d_{31}$ is much higher than that of $d_{33}$ [39,66,67].

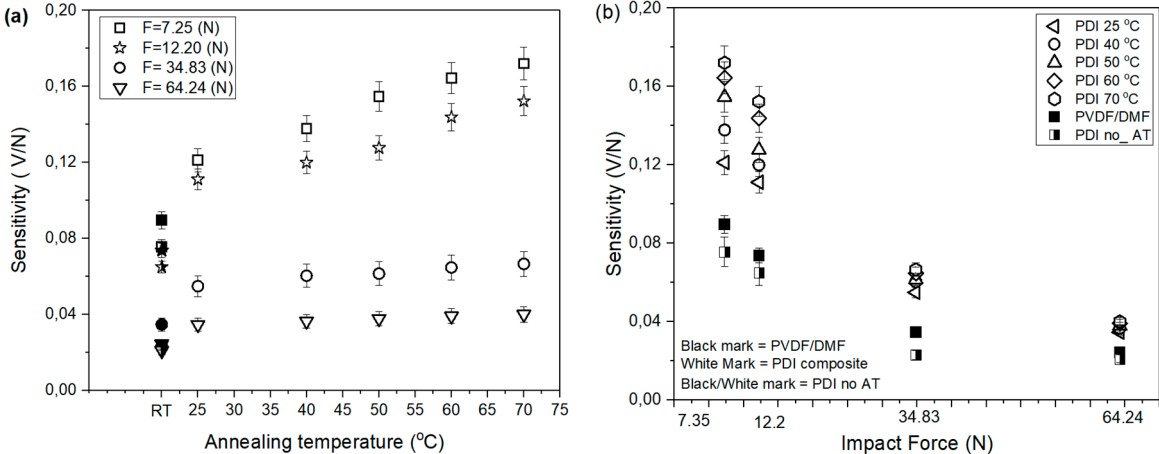

**Figure 8.** (**a**) Sensitivity ($d_{33}$) vs. annealing temperature; (**b**) sensitivity vs. impact force.

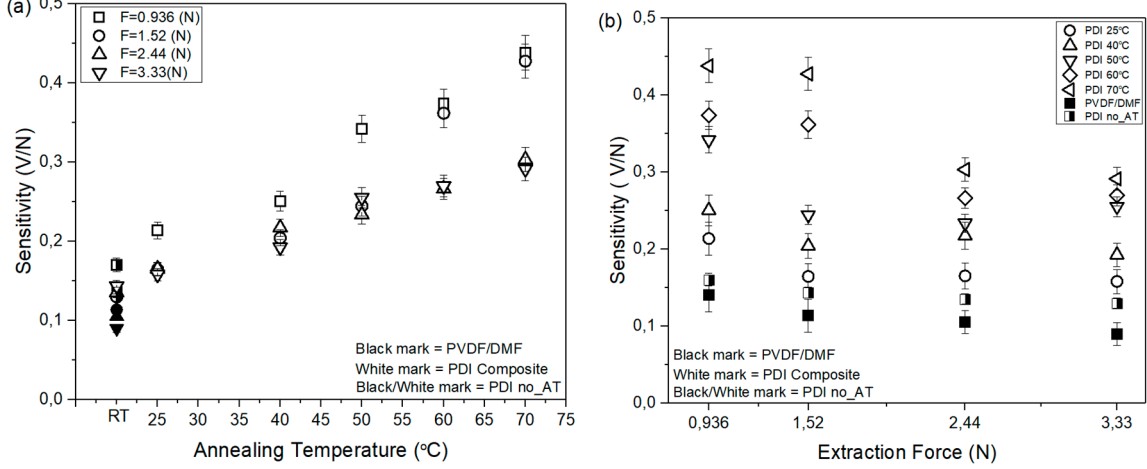

**Figure 9.** (**a**) Sensitivity ($d_{31}$) vs. annealing temperature; (**b**) sensitivity vs. extraction force.

### 3.5. Measurement of Piezoelectric Coefficient ($d_{33}$ and $d_{31}$)

The piezoelectric charge constant $d_{33}$ is highly affected by the amount of β-phase [38,68]. A commercial $d_{33}$ m (Piezoelectric $d_{33}$ m Model SINOCERA YE2730) was used to measure the piezoelectric coefficient $d_{33}$ of the PDI composite film as shown in Figure 10 with expression of error bars. As seen, the piezoelectric coefficient is almost linearly increasing as the temperature of hot water is increased. A maximum piezoelectric coefficient $d_{33}$ value of 24 pC/N was obtained for the PDI composite film after immersion in hot water at 70 °C for 2 h, which is close to the standard 27 pC/N [36]. The piezoelectric properties $d_{31}$ was calculated by [39,66,67].

$$d_{31} = \varepsilon_r.\varepsilon_o.g_{31} = (\varepsilon_r.\varepsilon_o).\frac{V}{t.\sigma} = (\varepsilon_r.\varepsilon_o).\frac{Vw}{F_e} \tag{5}$$

where $g_{31}$ is the piezoelectric voltage coefficient; σ ($=F_e/tw$) is the tensile stress; $F_e$ is extraction force measured by the commercial force sensor (PCB piezoelectric, model 208c01); $t$ and $w$ are the thickness and width of PVDF film, respectively. Same as the $d_{33}$ results, the piezoelectric coefficient $d_{31}$ shown in Figure 10 is nearly proportional to temperature increase of hot water. Also, the maximum $d_{31}$ occurs at a temperature of 70 °C. As usual, the piezoelectric coefficient $d_{33}$ is much larger than $d_{31}$.

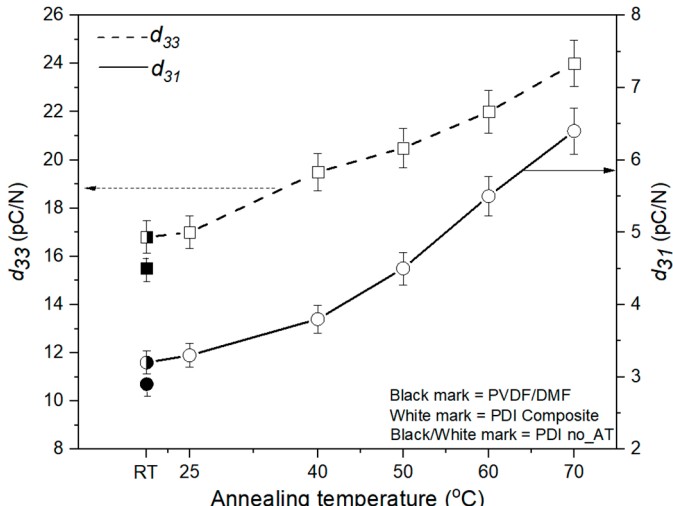

**Figure 10.** Piezoelectric coefficient $d_{33}$ and $d_{31}$ for different annealing water temperature.

### 3.6. Surface Morphology

Figure 11 shows the surface morphology and microstructure images provided by SEM. Surface morphology of the crystallized PVDF film samples was observed from the top surface of the film samples, i.e., opposite to the surface in contact with the glass substrate (bottom surface). Figure 11a shows the granular structure of the top surface of PVDF/DMF film in α phase, which shows a spherulitic structure with porosity (labeled by blue color arrow) when crystalized from solution or from the melt. The solvent properties, solution casting, film forming process, evaporation rate, and film thickness also influences the porosity. The presence of pores on the surface of the film will hamper the deposition of electrode and reduce the electrical response that makes the later poling process difficult.

Surface micrographs of the PDI no_AT film is shown in Figure 11b. The crystalline morphology structure of the film sample is significantly influenced by the addition of IL in polymer matrix. Spherulitic microstructure was found existing in film sample. As seen in the images, numerous numbers of different shape and size of IL particles are randomly distributed on the surface film, which implies not all the IL are mixed into the polymer matrix. However, the addition of IL into the polymer matrix would apparently reduce porosity.

For further investigation, selected examples of the surface morphologies of the PDI composite films prepared by immersion in water at temperature of 25 °C and 70 °C are shown in Figure 11c,d, respectively. As can be seen, the surface of PDI composite films displays relatively smooth surface morphology indicating homogenous distribution in the composition and good compatibility with polymer matrix [69]. In Figure 11c, it can be observed that the spherulitic microstructure structure is similar to that of the PDI no_AT shown in Figure 11b. However, the majority of the IL were removed while immersing in warm water. Under such condition, the results may come from not high enough solubility of IL and weak interaction with the PVDF chains. While increasing hot water temperature to 70 °C, the spherulites size become smaller and more compact with only miniature-sized porous structure on the surface of the film sample as shown in Figure 11d. It indicates that IL as an anion has strong electrostatic connection with the PVDF polymer chain and would raise a large amount of very small spherulites. The strong electrostatic interaction between the dipolar moment of PVDF matrix and the IL would lead to an increase of β-phase content.



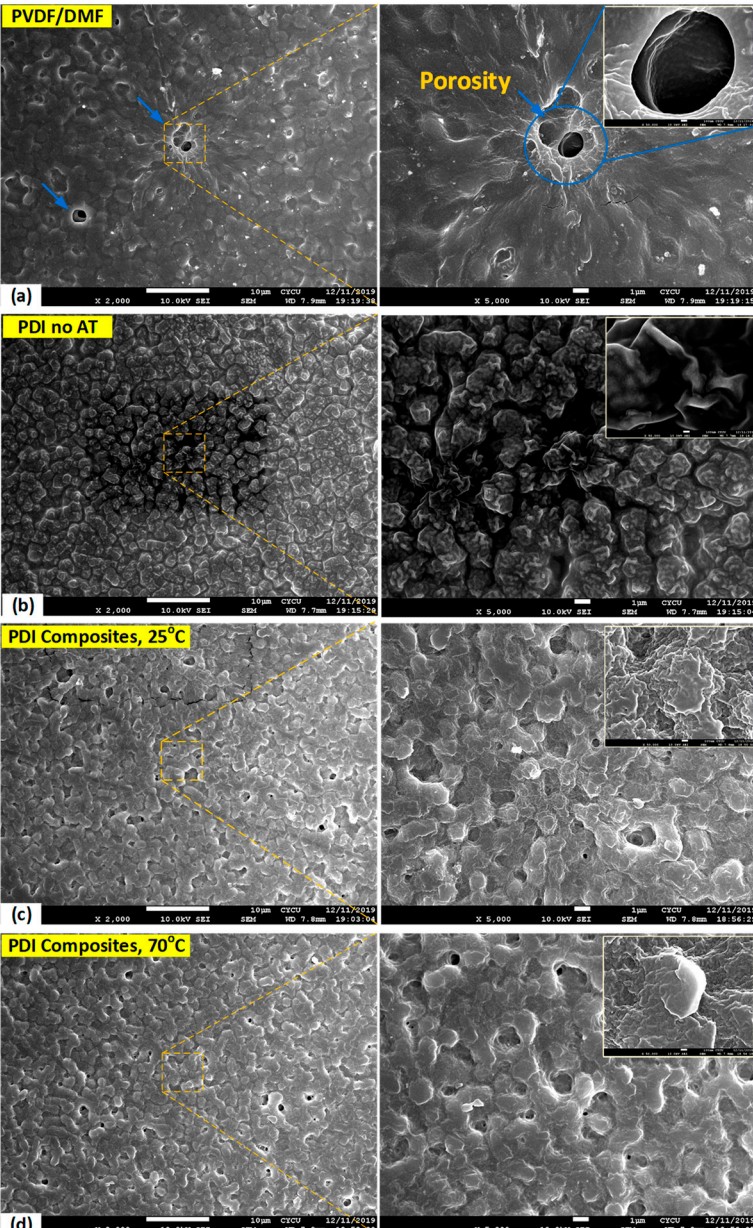

**Figure 11.** SEM images of (**a**) PVDF/DMF, (**b**) PDI no_AT, and (**c**,**d**) PDI annealed at 25 °C and 70 °C.

### 3.7. Atomic Force Microscopy (AFM)

In Figure 12, three-dimensional surface image morphology carried out by AFM was used to observe the surface topography of PDI composite films. AFM imaging was examined in contact mode in which the tip is placed in contact with the surface of the PDI composite film at room temperature. Figure 12a presents the three-dimensional topography images of the fabricated PVDF/DFM film. The dark-color solid line (—) and bright-color dotted line (····) domains represent the valley (depth) and peak (height) on the surface of samples respectively [39]. Figure 12b–f shows the surface morphology of the PDI composite films that were immersed in hot water at different annealing temperature. The surface morphology image of the sample becomes smoother, which is mainly due to the enhanced interaction between the IL and the amorphous phase of the matrix polymer as well as the fast washout of IL in high annealing temperature especially [70]. Figure 12f shows the surface of the PDI composite film is smoother with small pores as well as small diameter of spherulites after immersion in hot

water at 70 °C for 2 h. Furthermore, particle spherulites were not found on the surface morphology, indicating that more particle aggregation of IL was removed on the hot water.

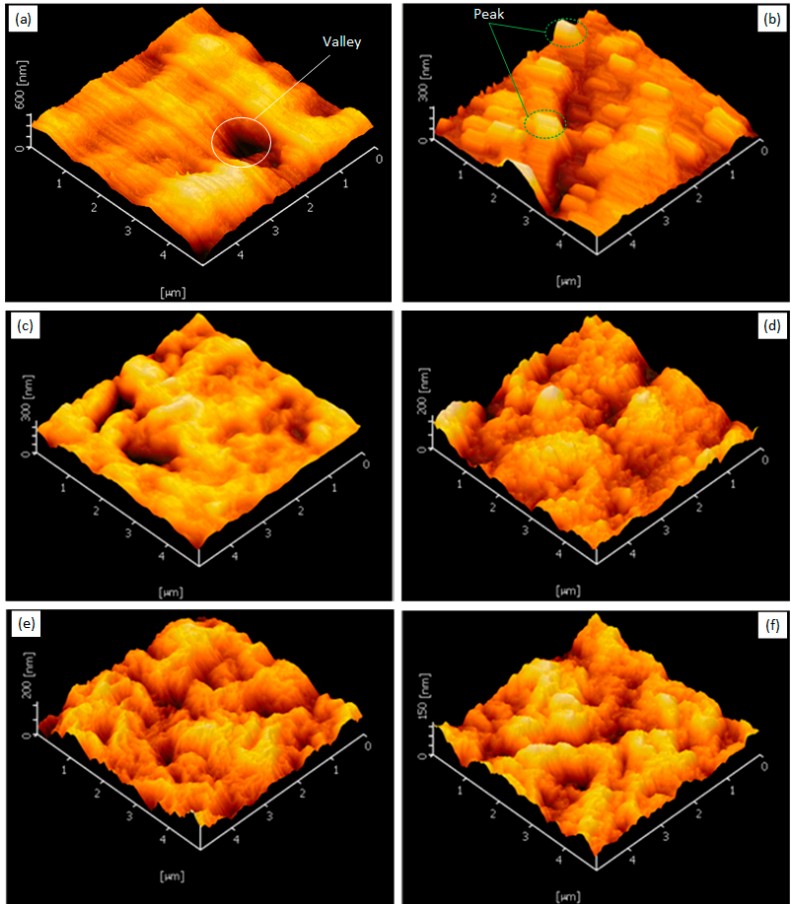

**Figure 12.** AFM Images (**a**) PVDF/DMF film, PDI composite film at (**b**) 25 °C (**c**) 40 °C (**d**) 50 °C (**e**) 60 °C (**f**) 70 °C

In general, surface roughness data give valuable information of surface morphology [71]. A large number of peaks and valleys appeared in the AFM image, which would affect the surface roughness values significantly [72]. Values of surface roughness of the PVDF/DMF and PDI composite films were directly read on the AFM, respectively, and shown in Figure 13 with the expression of error bars. From the surface roughness analysis, it is seen that the PVDF/DMF film has large-scale roughness compared to the PDI composite films mainly because of the addition of IL. Also, the surface roughness is decreased as annealing water temperature is increased. In particular, the PDI composite film immersed in hot water at a temperature of 70 °C would result in lower roughness than the other treatments. Both the AFM and SEM results are in a good agreement and consistency. It implies that significant synergetic effects on both the formation of polar crystal PVDF by melt crystallization and by annealing treatment in hot water is instrumental to complete the β-phase crystallization and reduce porous pure β-PVDF film effectively [36,47].

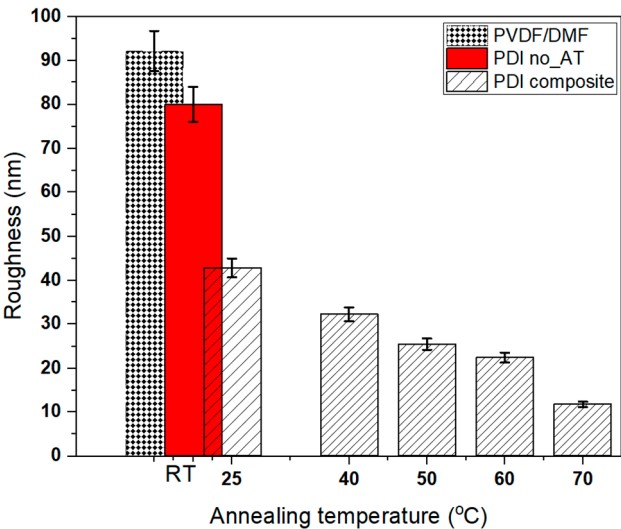

**Figure 13.** Surface roughness of PVDF samples in Figure 12.

## 4. Conclusions

In this study, PVDF film prepared by using the solution-casting method mixed with solvent DMF and filler ionic liquid is investigated. Besides that, after casting and evaporation, the composite film is immersed into hot water for annealing treatment. Characteristic analyses were carried out by means of FTIR, XRD, SEM, and AFM to examine the effect of ionic liquid (IL) in PVDF/DMF and annealing treatment at different temperature. As found in the experiment, addition of IL in PVDF/DMF will facilitate the interaction between the $CF_2$ and cationic ions. The ion-dipole effectively interacts with the ionic liquid and PVDF chains to promote the formation of trans sequence in the melt [36]. Transformation from α-phase to β-phase is therefore enhanced, and piezoelectric properties, degree of crystallinity, and surface roughness is improved. Moreover, PVDF composite films immersed in hot water to perform annealing treatment can not only increase the interaction with the amorphous region in the polymer matrix and the nonpolar crystal in PVDF, but also accelerate IL removal. As hot water temperature is increased, but not beyond 70 °C, more reverse phase transformation happens to gain more β-phase fraction, and more IL is observed to remove out faster, which leads to reduction of the porosity of the PVDF film.

Also, IL is rarely seen in SEM and smoother surface is seen in AFM. The proposed solution casting method with the addition of IL and the subsequent annealing treatment can complete the fabrication of PVDF film directly. Without using complex stretching or hot press method, the PVDF film fabricated by the proposed simple way can provide satisfactory performance that will be contributory and influential to the later poling.

**Author Contributions:** Conception, wrote the paper, analyzed the data, Y.T.; wrote the paper, analyzed the data, conceived the experiments, collected data, S.S.; interpreted the data, contributed writing the manuscript, and experiments, B.N.; performed the experiments and collected data, S.K. and Y.A. All authors have read and agreed to the published version of the manuscript.

**Funding:** This research was funded by Ministry of Science and Technology (MOST), grant number 104-2221-E-033-011" and 108-2221-E-033-044, and sponsored by Sound-wide Technology Corp.

**Acknowledgments:** The authors gratefully acknowledge the financial support provided by Ministry of Science and Technology (MOST) a grant number 104-2221-E-033-011" and 108-2221-E-033-044, and sponsored by Sound-wide Technology Corp.

**Conflicts of Interest:** All authors declare no conflict of interest.

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
