# Peer review of "Using Annealing Treatment on Fabrication Ionic Liquid-Based PVDF Films"

_coatings, doi:10.3390/coatings10010044_

Round 1
Reviewer 1 Report
Manuscript number: coatings-666895
Manuscript title: Using Annealing Treatment on Fabrication Ionic Liquid Based PVDF Films
The authors investigated PVDF thin film using a solution casting process. The manuscript gives an overview of methodology and performance for the sensor and the paper is well organized and it will be interesting for lots of readers. My recommendation would be to accept the paper with further revisions as I have the following concerns:
Authors should also note that other piezoelectric materials (especially P(VDF-TrFE)) are being developed all over the world which also has the potential to convert and generate high output voltage response. This then brings forth the following question: how is PVDF described in this study superior to the other piezoelectric materials (e.g. P(VDF-TrFE))?
It would be better if authors give various piezoelectric materials and its applications help reader focus in the very different experiments whose results are reported. Add relevant references on piezoelectric materials and applications, like the following:
Applied Sciences 8(2), 213, 2018.
Polymers 10(4), 364, 2018.
Sensors 19(6), 1404, 2019.
Provide a more detailed explanation in the Introduction section for the readers while not only giving more information about other materials, performance, and examples from other literature, but also adding novelty of this work. This is too simple now. (What are the advantage and merit of this device compared to other research?)
Provide more detailed explanation in the experimental and characterization sections. This is also too simple.
Provide bigger and clear x-axis (numbers) in Figures 2 and 4. The numbers in the figure results are not easy to read.
In Figures 5 and 6, it looks like the output comes from only an impact force. I would like to see the results with multiple input forces (at least 5 cycles/each input).
In figure 10, the quality of SEM images are bad. Provide them with better quality and compare them at the same magnification. It looks like their all magnifications are different. Also, insert the scale bars.
Reviewer 2 Report
The method proposed by Yung Ting et al. is a faster and easier method to fabricate high-quality PVDF thin films. This paper is of interest to readers of "Coating" and is listed below and recommended for acceptance after revision of the manuscript.
1. The abstract and discussion are very qualitative. It is recommended to describe them quantitatively.
2. In the IR spectra, the wavenumber at 1072 cm-1 was a shift by increasing the temperature. Please mentioned the reason.
3. The peaks of γ-phase at 431 and 482 cm-1 did not appear in the figure. Please show it in Fig. 2.
4. In Fig.4, the 2θ value of the x-axis described comma. Please using the period.
5. The immersion water temperature is an essential factor of annealing treatment would affect the fraction of β-phase. And, the piezoelectric coefficient d31 shown is nearly proportional to the temperature increase of hot water. How about immersion water temperature at 80°C? Is this trend up to 70 ° C?
Reviewer 3 Report
The authors mentioned in introduction section that the proposed process of producing large amount of b-phase is unlikely to have significant fraction of γ-phase, due to the required extremely high temperature to produce large γ-phase content [11, 14]. However, the FTIR and XRD measurements show γ-phase content. Can authors clarify that contradiction?
The description of SEM and AFM techniques were explained in Results and Discussion part, should be moved to Materials and Methods.
Please avoid basic expressions, such as (line 216) Sensitivity is a key parameter to sensors [47]!
Round 2
Reviewer 1 Report
Authors have revised the manuscript well based on my comments.